# Life Satisfaction and Academic Engagement in Chileans Undergraduate Students of the University of Atacama

**DOI:** 10.3390/ijerph192416877

**Published:** 2022-12-15

**Authors:** Carmen Burgos-Videla, Ricardo Jorquera-Gutiérrez, Eloy López-Meneses, Cesar Bernal

**Affiliations:** 1Department of Primary Education, Institute for Research in Social Sciences and Education, Vice-Rectory for Research and Postgraduate, University of Atacama, Copiapó 1530000, Chile; 2Department of Psychology, University of Atacama, Copiapó 1531772, Chile; 3Department of Education and Social Psychology, Pablo de Olavide University, 41013 Seville, Spain; 4Ecotec University, Km 13.5 Vía Samborondón, Samborondón 092302, Guayas, Ecuador; 5Department of Education Sciences, Language, Culture and Arts, Rey Juan Carlos University, 28032 Madrid, Spain

**Keywords:** digital competencies, latent class analysis, latent class models, technology consumption, data analysis

## Abstract

The growing problem of mental health in the university population, as a consequence of the COVID-19 pandemic, has generated the need to consider positive variables to address this situation. Life satisfaction and academic engagement are two constructs that emerge as conceptual tools oriented in this direction. The present study sought to describe the effect of academic engagement on life satisfaction in a sample of Chilean university students. A cross-sectional co-relational design was used. A total of 370 university students participated, 72.4% female and 27.6% male, aged beitive effect of engagement on life satisfaction was demonstrated, where the dimensions vigor (β = 0.462; *p* < 0.01) and dedication (β = 0.465; *p* < 0.01) acted as significant predictors (χ^2^ = 87.077, gl = 32, *p* < 0.01; χ^2^/gl = 2.721; CFI = 975; TLI = 0.964; RMSEA = 0.068). The proposed model showed factorial invariance according to sex. The usefulness of employing these constructs as a way to manage the well-being and mental health of students in university institutions is discussed.

## 1. Introduction

The right to education is one of the fundamental human rights. As a result, international instruments that seek to safeguard its full exercise have emerged. Knowing that the right to education enables progress in achieving sustained well-being, education and its processes become fundamental and a focus of national and international concerns. The right to education addresses different dimensions: access and permanence, as well as the quality of education, together with coexistence and good treatment within the educational system. Each of these dimensions are closely related and are determinant for the proper exercise of the right to education. In view of the above, the right to quality education is essential; therefore, it is necessary to consider the ways in which educational institutions take charge of its exercise, and how quality is implemented in order to make evident the opportunity to educate in healthy coexistence, in spaces that promote well-being and satisfaction in life.

In Chile, there is an education system that is unequal, and this goes beyond the identity of students and teachers, who show dissatisfaction with the development of training processes in different structural and psycho-social senses [1]. As a result, the emergence of research that includes psychological aspects is increasing. During the last twenty years, research on the educational issue of satisfaction has considered the subject with interest, observing and analyzing it from the perspective of Positive Educational Psychology, i.e., there has been an increased concern in the factors involved in the positive behavior and psychological well-being of the subjects [2,3]. In this sense, studies are advancing to respond to the problems of performance, motivation, and others such as student desertion. In this context, academic engagement [4] would be a potential aid in the presence of psychosocial problems, because from theory, it promotes learning and performance in a conscious way [5]. According to the research of different studies, it is difficult to find advances in studies that address the association between life satisfaction and academic engagement in Latin America. Academic engagement can be seen as the result of a successful adaptation to the environment with a good academic performance [6] However, the influence of life satisfaction on academic engagement has not been determined.

The search for new theoretical references to define what is required for the achievement of educational quality recognizes this construct in the opportunity to advance in the understanding and continuous improvement of complex scenarios, such as life and education during the pandemic.

It is known that the arrival of SARS-CoV-2 caused the decline of the improvement processes to reach educational quality and raised the need to search for new ways of teaching [7]; more importantly, we should consider that the student body has different economic possibilities as well as the impact of the digital gap in learning, since not all students can access technology, due to their geographical location and vulnerability indexes. This situation exacerbates the feeling of frustration and demotivation of the people involved, lowering life satisfaction and, therefore, the academic engagement.

As a result of the mentioned context, a large number of students drop out of school, stay at home taking care of their grandparents or relatives, are absent from the processes of their training because they live in rural areas and are far away from institutions, sometimes without electricity and internet connection [4,8]. Due to the complexity of the pandemic context, research based on positive psychology began to be developed, focusing on the strengths of each person to deal with crises and stress situations, such as the COVID-19 pandemic.

### 1.1. Academic Engagement

Studies show that Academic Engagement (hereinafter AE) [9] is essential to increase learning and obtain better performance and well-being of the students. Being a polysemic construct, it is complex to define. However, different authors refer to its indicators and dimensions that constitute its field of action. The appearance of relationships between these increases the need to make visible its exercise in the student’s life. According to some authors, in order to define AE, we must pay attention to vigor, dedication and concentration [10]. Vigor is the maximum energy put into the task to be performed. For some authors, dedication would be essential for motivation to continue [11], and concentration would be related to enthusiasm and the desire to become involved in the task.

One of the most powerful theoretical models proposed [12], is the one that evidences the structure of academic engagement. It was proposed by Salanova et al. [13] as a motivation state that is generated through the association of the dimensions of engagement, which are vigor, dedication, and absorption. Vigor, as already mentioned, is related to the desire to strive for a task and refers to the energetic impulse that helps persistence and to a powerful desire to put effort into studies.

Dedication is characterized by the significance of what is being done. It concretely shows the identification with a career or with what is being studied. Meanwhile, absorption is related to the possibility of involvement in learning, considering that this is generated from interest and motivation. In the search for the dimensions of engagement, there is consensus among other authors who established that the dimensions of vigor and dedication are the heart of EA [14,15]; activation with direction, being part of motivation, would become the common denominator in the dimensions of engagement [16].

According to Ref. [17], in order to develop certain behaviors, activation with an energy level in accordance with the task and the achievement of its objectives is necessary. This is associated with vigor and motivation in the direction of a certain object. Thus, it is explained that motivation is related to vigor and will be one of the behaviors that are used to achieve goals. [18,19]. In this understanding, we can speak of mental endurance when working and studying. The dedication to achieve a goal provides information on the involvement, while enthusiasm provides information on the purposes, and absorption, as mentioned above, provides information on concentration. Consequently, to achieve academic success, it is necessary that the student is involved in the task with motivation and concentration, while having a positive appreciation of the work being accomplished [20].

### 1.2. Life Satisfaction

Life satisfaction is a key psychological dimension of well-being. Life satisfaction is defined as the judgment that a person makes of their life in global terms [21]. Life satisfaction represents a total analysis of life, a type of evaluation which is considered as a broad value judgment that a person makes of their life. For this, the person observes and analyzes all the factors and criteria that they consider important, always from their point of view, from the order of priority that they consign to these criteria [22]. This way of evaluating life considers the comparison with standards and expectations that the person has built from their perception of life and their experiences, and has come to represent, both in their thinking and in concrete reality [23].

There are studies that address perceived personal discrimination, which is defined as a perception of experiences that make one feel rejected, excluded, and discriminated against by a group, affecting life satisfaction [24]. Other studies address the empirical evidence on the negative relationship between psychological expressions of discomfort or well-being with respect to perceived prejudice, i.e., that life satisfaction is affected by perceived prejudice [25]. On the other hand, there are studies that highlight that self-esteem is one of the relevant predictors of life satisfaction; however, it has been shown that its incidence is greater in individualistic cultures than in cultures that develop collectively [12].

Life satisfaction is considered an indicator of quality of life [26], and the latter implies living in acceptable conditions that allow a person’s fulfillment in accordance with ideological values and a subjective life experience that is satisfactory in accordance with the life experience the person has. With respect to life satisfaction, two components are clearly identified: an emotional component, in which there are positive and negative affects [27], and a cognitive component, named life satisfaction [28]. The theories devoted to the study of psychological well-being have not been able to explain what provokes, causes, and results in the effect of the construct of satisfaction. Thus, there is a gap regarding the explanation of satisfaction as a set of happy moments in the course of the existence and the understanding of the well-being of people as a disposition towards life that is general and related to personality, ensuring satisfaction in all the dimensions of what is considered vital.

### 1.3. University Context

The university context went from being considered a space where students who entered could remain because they were emotionally stable—so scholars and researchers could collect information—to a worrying space [29] where there is desertion, illness, mental health problems, preoccupation with suicides, and generalized demotivation.

According to the National Institute of Mental Health of the United States, one of the ten most important causes that affect the economy of societies are mental health diseases. In view of this, it is important to note that the university context is not out of this determination; indeed, it is considered a prone scenario to avoid the shortcomings regarding disorders and mental health. This has led the World Health Organization (WHO) to make an effort to address the issue as a priority in research, such as in programs that help prevention and collaborate with the treatment of mental illnesses: “Adolescence is considered when a person is between 10 and 19 years of age. They are a healthy group; however, many die prematurely due to accidents, suicide, violence, pregnancy-related complications, and preventable or treatable diseases” [30].

Higher education students have experiences that include important risk factors with the absence of protective factors from their homes and their student and personal lives before entering higher education [31]. Therefore, there is a set of biopsychosocial and cultural variables that accentuate success or failure in the student stage, for example, the competitiveness that exists with peers or subjects that have earned places of acceptance, school bullying, conflicts, friendship and love affairs, among others. Faced with this reality, university students will protect themselves with different resources of individual or collective management, will be defensive of stressful moments, facing them sometimes with anger and rage. It is important to point out that beyond the possibility of protecting themselves, they find close circles in the university, and sometimes, the necessary support in the family; in the same way, there are new alternatives of care and protection, such as alternative therapies to overcome emotional imbalances. Thus, it is evident that sometimes, university life is not synonymous of well-being and personal satisfaction [32].

In the population of university students, there is an increase of psychosomatic problems, which are problems without an organic cause but with an impact on a young person’s behavior and psychological functioning with respect to the environment and themselves [33].

Somatic symptom disorder in adolescents is assiduous and interferes with their functioning [34]. Anxiety is presented by adolescents, as evidence of anxious sensitivity and negative affect, because they feel pressured in their environment. Anxiety appears as an element that has an impact on the development of positive life satisfaction [35].

Anxious sensitivity (hereinafter, AS) also influences life satisfaction and constitutes an important predisposing factor for the development of anxiety disorders, where fear and fear of making mistakes in different matters becomes very important and a trigger for certain psychological characteristics that go against a healthy academic development [36]. According to current research, AS is shaped by a general factor and three interconnected dimensions: physical, cognitive, and social [37].

On the other hand, there are studies that show that most adolescents show satisfaction with relevant aspects of their lives [38]; however, these studies are not conclusive, because life satisfaction is influenced and shaped by other important variables, such as age, gender, and geographic location [39]. The difference with respect to sex is notorious, because men show a higher level of satisfaction than women [40]. Men express greater satisfaction with their personal attributes and achievements in their work and family life, while women show greater satisfaction with their personal relationships. [41] In a study dedicated to the validation of a stress-coping scale, it was concluded that, according to gender, the prevalence of stress is perceived to be higher in women than in men.

Whatever the situation is, when a young person faces moments that they cannot overcome using their psychological or protection tools related to the support of the family or networks of friends, they are subject to negative stress that hinders their achievement of life satisfaction [42]. Another issue that becomes apparent to achieve life satisfaction will be the material lack [43,44].

In order to advance in the present research, it is necessary to consider the above-mentioned points. In this sense, the following table shows the components of engagement. It is observed in Table 1.

Taking into consideration what has been explained with respect to engagement, life satisfaction and the university context, the present study aimed to describe the effect of academic engagement on life satisfaction in Chilean university students from Copiapó to reveal the state of this association and to be able to offer spaces for the achievement of quality education. It is observed in Table 2.

## 2. Methodology and Instrument

The design used in this research was consistent with the objective of the study, which was to describe the effect of academic engagement on life satisfaction in Chilean university students from Copiapó. For this purpose, non-experimental research of a cross-sectional correlational type was considered. It should be noted that in non-experimental research there is no deliberate manipulation of the variables, and the phenomena are only observed in their natural environment and then analyzed [48]. On the other hand, the present research was cross-sectional–correlational, since a description of the study variables was sought by establishing the relationship between them at a single point in time.

The stages were mainly based on a data analysis process and included the following:Data preparation: at this stage, a descriptive analysis of the data was carried out, in which basic statistics were obtained. Then, correlational analyses were carried out, and the proposed predictive model was verified using the AMOS 21 program.Model generation and evaluation: the adjustment of the proposed structural model was verified, considering life satisfaction as a criterion variable and engagement and its dimensions as predictors. At the same time, the factorial invariance of the hypothesized model was estimated according to sex.Analysis of results: the results obtained from the techniques described in the data analysis section were analyzed.

### 2.1. Data Collection Instruments

Academic Engagement. The instrument used was the Spanish adaptation of the Academic Engagement Scale UWES-S9. The scale is composed of 9 items divided into three subscales with adequate Cronbach’s Alpha levels: [49] vigor (3 items, α = 0.81), absorption (3 items, α = 0.73), and dedication to studies (3 items, α = 0.82). It is answered through a 7-option Likert scale ranging from 0 = not at all to 6 = every day. In a study conducted in Chile by [50], the UWES 9S showed McDonald’s ω reliability indicators between 0.67 and 0.83 in its three factors, showing support for its structure and demonstrating its invariance according to sex at different levels of study.

Satisfaction with life. The Satisfaction with Life Scale (SWLS) [51,52] was used, which is composed of 5 items that assess people’s overall judgment of their satisfaction with their own life. It has a 7-point Likert-type response system. The psychometric properties reported in samples of Chilean students have been positive, showing adequate internal consistency (α = 0.87) and a unifactorial structure [53].

### 2.2. Data Analysis

First, the descriptive statistics of the variables were estimated, specifically, means, standard deviations, skewness, and kurtosis. Cronbach’s alpha was also estimated for each of the scales used to evaluate these variables and their dimensions.

The correlational analyses were carried out using the Pearson correlation coefficient. Subsequently, the goodness-of-fit of the hypothetical structural model was verified by validating the linear regression values of each of the variables of the model through structural equation modeling (SEM). The goodness-of-fit estimation was carried out using the maximum likelihood method. The rates considered were chi^2^ chi^2^gl, the Comparative Fit Index (CFI), and the Tucker Lewis Index (TLI). In addition, the Root-Mean-Square Error of Approximation (RMSEA) was analyzed. A non-significant chi2, a chi2/gl value lower than 3 and values higher than 0.90 in CFI and TLI were considered as good-fit indicators. The root-mean-squared residuals of approximation (RMSEA) were also evaluated, considering values lower than 0.08 [54]. Finally, the factorial invariance of the instrument according to sex (women and men) was estimated. For this purpose, the adjustment indicators used were the same as those used in the previous CFA, describing and comparing sequentially the configural, metric, scalar, and strict invariance of the instrument, observing that the variations in the CFI values between each procedure were less than 0.010 and that the variations in RMSEA were less than 0.015 [55,56].

The correlational procedures were performed using the statistical package IBM SPSS 22.0 for Windows. The estimation of the goodness-of-fit of the structural model was performed using the AMOS 21 program for Windows.

### 2.3. Population and Sample

The study sample was non-probabilistic and consisted of 370 students from a university in northern Chile, 72.4% of whom were women, and 27.6% were men. The ages of the participants ranged from 17 to 39 years (M = 21.34; SD: 3.2). All of them voluntarily agreed to answer the evaluation instruments, after reading and signing an informed consent form. The inclusion criteria considered were to be enrolled in a university course of study and the voluntary participation of each one of them.

In relation to the inclusion criteria, this study considered those students enrolled in a degree program at the university where this research was developed, who were attending a degree program of eight or more semesters from the Schools of Humanities and Education, Medicine, Health Sciences, Engineering and Technology. The exclusion criteria regarded those students who were not enrolled in the university of this research, who followed a program of less than eight semesters, or who belonged to a different faculty from those considered in the study.

The students were contacted through their institutional e-mail addresses; they were sent an informed consent form which explained the objectives of the research, the voluntary nature of participating in it, and the safeguarding and use of the information obtained and contained the contact information of the responsible researchers. This consent form, as well as the instrument, was accessed through a Google Form link, which allowed the students to answer remotely. Each student could answer the questionnaires only once. A total of 414 responses were received; all subjects who did not meet the established inclusion criteria were excluded. As seen in Table 3.

## 3. Results

The results showed that life satisfaction correlated positively and significantly with the three dimensions of academic engagement. These relationships ranged from 0.454 to 0.508, all of which were significant at the *p* < 0.01 level. Let’s look at Table 4 below.

Table 5 describes the mean comparisons of the variables studied according to sex and educational level. According to sex, no significant differences were observed in life satisfaction (t = 0.41, *p* = 0.68), vigor (t = −1.04, *p* = 0.30), dedication (t = −0.80, *p* = 0.43), and absorption (t = −1.86, *p* = 0.06).

When life satisfaction was compared according to the level of studies, there was no significant difference between the groups evaluated (F = 0.30, *p* = 0.88). However, at the level of the engagement dimensions, significant differences were observed in the dimensions of vigor (F = 5.35; *p* < 0.001; ηp^2^ = 0.06), dedication (F = 5.59; *p* < 0.001; ηp^2^ = 0.06), and absorption (F = 3.29; *p* = 0.011; ηp^2^ = 0.04). Performing a post hoc Tukey test, in the case of vigor, significant differences (*p* < 0.05) were observed between the first-year group and the second-, third-, fourth-, and fifth-year groups. For dedication, significant differences (*p* < 0.05) were observed between the first-year group and the third-, fourth-, and fifth-year groups. For absorption, a significant difference was only observed between the first- and the fourth-year groups (*p* < 0.05). In all these cases, the highest levels of vigor, dedication, and absorption were observed in the first-year group.

A structural model was subjected to empirical testing in an attempt to verify the predictive power of academic commitment on life satisfaction (see Figure 1). In this regard, it was observed that vigor (β = 0.462; *p* < 0.01) and dedication (β = 0.465; *p* < 0.01) significantly predicted life satisfaction. On the other hand, absorption did not prove to be a variable that significantly predicted life satisfaction (β = −0.301; *p* = 0.208). The predictor variables were able to explain 36% of the variance in life satisfaction. The structural model showed acceptable fit values (χ^2^ = 263.48, gl = 71, *p* < 0.01; χ^2^/gl = 3.711; CFI = 947; TLI = 0.932; RMSEA = 0.086). However, it is necessary to point out that when eliminating the absorption variable from the model, all the fit indicators improved (χ^2^ = 87.077, gl = 32, *p* < 0.01; χ^2^/gl = 2.721; CFI = 975; TLI = 0.964; RMSEA = 0.068).

Table 6 shows the analysis of the factorial invariance of the proposed model according to sex, showing that the data support that the structural model is invariant for the group of women and men. The configural invariance was analyzed, showing adequate values (CFI = 0.934; TLI = 0.915; RMSEA = 0.068). Subsequently, metric invariance was evaluated, showing favorable values (CFI = 0.934; TLI = 0.921; RMSEA = 0.066). When comparing the results of the metric and configural invariance, no significant differences were observed (ΔCFI = 0.000 and ΔRMSEA = 0.002). The results allowed us to establish that the factor loadings were invariant between the two groups. Thirdly, scalar invariance was evaluated. The results showed adequate fit indices (CFI = 0.934; TLI = 0.925; RMSEA = 0.064). When comparing the results of metric and scalar invariance, no significant changes were found (ΔCFI = 0.000 and ΔRMSEA = 0.002), which allowed us to point out that the intercepts were invariant in the male and female groups. Finally, strict invariance was analyzed, and adequate fit indices were found (CFI = 0.934; TLI = 0.932; RMSEA = 0.061). The comparison of the scalar invariance adjustment indices showed adequate differences (ΔCFI = 0.000 and ΔRMSEA = 0.003), providing empirical support for strict invariance.

## 4. Discussion

The results provide a favorable background to confirm academic engagement as an antecedent of life satisfaction, which confirms the importance that the literature has shown, presenting it as an important predictor of academic performance and student well-being [9].

In particular, it was shown that higher levels of vigor and dedication would be important indicators in the prediction of a very relevant aspect of subjective well-being, i.e., life satisfaction. In this sense, on the one hand, high levels of a person’s energy, persistence, and desire to make every effort to succeed in their studies, and a high level of pride and identification with the career a person is pursuing would translate into higher levels of satisfaction with their own life. The relevance of these two dimensions has been considered in other studies where it has been pointed out that vigor and dedication are the heart of EA [14,15]. Consistent with the above, in the case of absorption, which refers to high levels of concentration towards studies, although it is a dimension related to life satisfaction, it would not necessarily be a predictor.

In the present study, no significant differences were observed in the variables studied according to sex. In the case of the life satisfaction variable, this contrasts with research in which significant differences have been found, with men having greater satisfaction with their lives [40]. Interpretatively, it could be presumed that the growing affectation of the mental health of the student population could be reaching a level where personal discomfort has been homogenized and with it also, in the opposite way, well-being. On the other hand, the absence of gender differences in academic engagement in Chilean university students has already been noted by other authors [50]. High levels of academic engagement in the first years of a career have been found in studies conducted with students of various careers in other countries [57,58]. However, different observations have also been reported, with students in their final years showing high levels of academic engagement [59,60]. Some lines of interpretation regarding the latter evidence point to the importance that professional internships have in the academic engagement of students in their last years, since these favor a more direct contact with the professional realities of the discipline [61]. At this point, it makes sense to distinguish that this study was conducted at a time when the students were taking classes and, in many cases, were also taking their professional internships, in a remote format, given the confinement policies resulting from the COVID-19 pandemic. For this reason, it is possible to assume that the students could not have an experience that would favor and enhance the development of their professional identities and a greater academic commitment in their professional practices, typical of higher levels of study.

The results showed higher levels of engagement in the first level of studies, compared to the higher levels. This would provide evidence that the levels of energy, identification, and concentration in the study tend to decrease as a student’s career progresses. This was also found in another study carried out in Chile in the same population.

Finally, the good fit of the proposed predictive model was highlighted. It also provided evidence of its structural invariance for men and women. Therefore, it is possible to point out that engagement would be an important predictor for university students, whether they are men or women.

A relevant consideration that can be inferred from these results is that it is possible to manage student well-being and, thus, mental health in the university environment through the management of academic engagement. In this regard, the generation of measures to promote academic engagement and the training of faculty and university managers in these issues should favor the improvement of mental health, a variable that has been greatly affected in the context of the COVID-19 pandemic.

## 5. Conclusions and Future Work

This study allowed verifying the impact of academic engagement on life satisfaction in Chilean university students. With the above, it can be confirmed that this is a relevant theoretical–conceptual tool for the promotion of mental health and the prevention of mental health issues in higher education institutions, based on a positive approach.

One of the limitations of the present study regards its sample, which was limited to only one higher education institution. Therefore, future studies should focus on the behavior of these variables in other higher education institutions, both public and private, and in other geographical areas of the country, in order to evaluate the possible generalization of the results to different institutional and academic realities. It also remains a challenge for further studies to observe the impact of its application in interventions in educational organizations.

## Figures and Tables

**Figure 1 ijerph-19-16877-f001:**
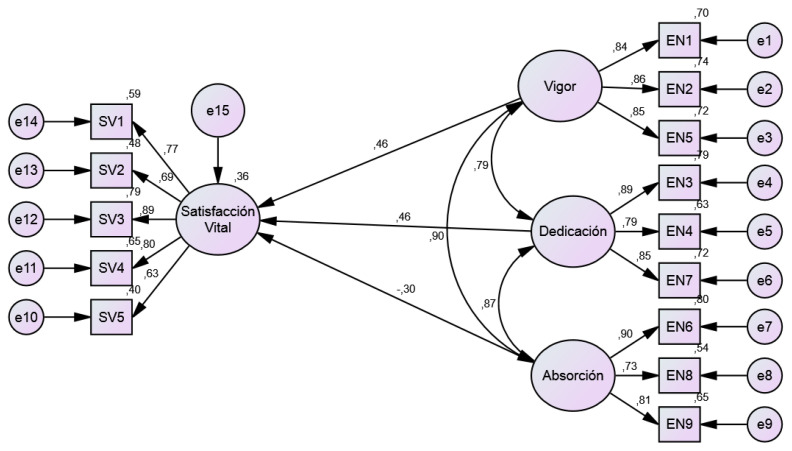
Structural Model of Life Satisfaction and Academic Engagement.

**Table 1 ijerph-19-16877-t001:** Engagement Components [1].

Vigor	Dedication	Absorption
High energy levels. Mental endurance during the tasks. Willingness to make an effort in the work. Persistency during difficulties.	High implication. Enthusiasm, dedication, pride. Meaning and inspiration.	Capacity to be absorbed and concentrated in a work or activity. Perception that the time passes very fast during work.

**Table 2 ijerph-19-16877-t002:** The benefits of engagement.

Demerouti et al. (2001); [2,3]	An engaged subject is proactive; possesses values aligned, high capacity for resilience and being engaged outside of work [45].
[4] and Demerouti et al. (2001)	An engaged subject presents increased levels of health, low levels of depression and nervous tension, and lower psychosomatic complaints [46].
[2] Bakker et al. (2011)	Engaged subjects show performance and quality of service [47].
[4]	Engaged subjects experience positive emotions, have a better state of health, generate their own resources, and transfer their engagement to others in their immediate environment.
Montoya and Moreno (2012) [3,5]	Engaged subjects feel more committed to the organization, have lower rates of absence, and do not show the intention to leave the organization; moreover, they experience positive emotions and enjoy a good mental and psychosomatic health (Tripiana and Llorens 2015).
	Subjects with high levels of engagement connect with their work, showing high levels of energy and mental stamina. They perceive their work as a challenge and show a high level of labor involvement (Ocampo et al. 2015).

**Table 3 ijerph-19-16877-t003:** Sample Distribution.

	Frequency	Percentage
**SEX**		
Male	102	27.6
Female	267	72.2
No Answer	1	0.3
**AGE**		
17 to 20 years old	183	49.5
21 to 24 years old	127	34.3
25 to 28 years old	48	13
29 to 32 years old	9	2.4
33 to 36 years old	1	0.3
37 years old or more	2	0.5
**CAREER LEVEL**		
1st year	97	26.2
2nd year	104	28.1
3rd year	76	20.5
4th year	43	11.6
5th year	44	11.9
Another	6	1.6

**Table 4 ijerph-19-16877-t004:** Descriptive Analysis, Reliability, and Correlations between the Study Variables.

	Mean	Standard Deviation	Skewness	Kurtosis	Cronbach’s Alpha	Life Satisfaction	Vigor	Dedication
Life satisfaction	21.103	6.885	−0.108	−0.786	0.862			
Vigor	8.081	4.965	0.033	−1.005	0.885	0.491 **		
Dedication	12.614	4.731	−0.836	−0.147	0.874	0.508 **	0.715 **	
Absorption	10.597	4.762	−0.453	−0.581	0.863	0.454 **	0.761 **	0.778 **

** Correlation is significative in the level 0.01 (bilateral).

**Table 5 ijerph-19-16877-t005:** Means, Standard Deviations, Student’s t, and ANOVA according to sex and educational level.

	Life Satisfaction	Vigor	Dedication	Absorption
**Sex**				
Men	21.35 (6.45)	7.69 (4.89)	12.33 (4.46)	9.88 (4.78)
Women	21.02 (7.02)	8.30 (4.99)	12.77 (4.77)	10.92 (4.72)
t	0.41	−1.04	−0.80	−1.86
*p*	0.68	0.30	0.43	0.06
**Study levels**				
First year	21.47 (7.29)	10.05 (4.91)	14.28 (4.02)	12.07 (4.22)
Second year	21.10 (7.02)	7.82 (5.10)	12.67 (4.72)	10.47 (4.81)
Third year	20.67 (6.70)	7.37 (4.84)	12.32 (4.68)	10.11 (4.88)
Fourth year	20.51 (6.50)	7.09 (4.16)	10.79 (5.12)	9.67 (5.27)
Fifth year	21.68 (6.25)	7.14 (4.70)	11.55 (4.62)	9.86 (4.51)
F	0.30	5.35	5.59	3.29
*p*	0.88	<0.001	<0.001	0.011
η^2^p	0	0.06	0.06	0.04

**Table 6 ijerph-19-16877-t006:** Factorial invariance of the model according to sex.

	X^2^	DF	X^2^/DF	CFI	TLI	RMSEA	∆CFI	∆RMSEA
Configural Invariance	384.642	142	2.709	0.934	0.915	0.068		
Metric Invariance	394.606	152	2.596	0.934	0.921	0.066	0	0.002
Scale Invariance	403.363	161	2.505	0.934	0.925	0.064	0	0.002
Strict Invariance	418.319	176	2.377	0.934	0.932	0.061	0	0.003

## Data Availability

They are kept in custody for one year.

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
