# Peer review of "Life Satisfaction and Academic Engagement in Chileans Undergraduate Students of the University of Atacama"

_ijerph, 2022, doi:10.3390/ijerph192416877_

Round 1
Reviewer 1 Report
The authors conducted a questionair study on unversity students to investigate the effect of sex and study levels on life satisfaction, vigor, dedication and absorption. The study found that sex has no impact on the three measured variables, while vigor, dedication and absorption showed significant difference between study level groups. The aim and scope of this research fits IJERPH, however, I cannot recomment publication due to the following major and minor points.
Major:
1. the authors emphasize Covid-19 in both the title and the discussion, however, I don't think this research is related to Covid-19. Although Covid-19 has been demonstrated to affect mental health by many other research, I didn't see how this research adds to the evidence. The research is not longitudinal so there is no comparison between pre- and post-pandemic mental health status using the same metric, and the effect cannot be inferred from the study levels as it coincide with the Covid-19 periods.
2. The other major conclusion lacks novelty. The decrease in vigor and dedication of undergraduate students by study levels have been shown in other studies, for example Morales-Rodríguez FM (2019). I suggest the authors dig dipper on the finding, such as try to explain why the first year students are most dedicated and vigorous, on what question did they show significant different answers to the other grades, or why the 4th year is the lowest.
Minor:
1. Please unify the way to interprete digits.
2. Tables and figures: please change the way the interprete numbers and texts align with the manuscript.
Author Response
|
corrector 1 |
|
|
observación |
Respuesta |
|
1. los autores enfatizan el Covid-19 tanto en el título como en la discusión, sin embargo, no creo que esta investigación esté relacionada con el Covid-19. Aunque muchas otras investigaciones han demostrado que el covid-19 afecta la salud mental, no vi cómo esta investigación se suma a la evidencia. La investigación no es longitudinal, por lo que no hay una comparación entre el estado de salud mental antes y después de la pandemia utilizando la misma métrica, y el efecto no se puede inferir de los niveles del estudio, ya que coincide con los períodos de COVID-19. |
La investigación se realizó en un contexto en el cual el estudio se realizó de forma remota dada la contingencia del covid 19. Sin embargo, no se trabaja con la pandemia del covid 19 como una variable independiente, sino más bien como un contexto particular que caracteriza de manera general y particular la forma como recibieron la docencia. El estudio es transversal y por ello, se entiende que es solo una fotografía del momento en el cual se realizó el levantamiento de información. |
|
2. La otra gran conclusión carece de novedad. La disminución del vigor y dedicación de los estudiantes de pregrado por niveles de estudio ha sido evidenciada en otros estudios, por ejemplo, Morales-Rodríguez FM (2019). Sugiero que los autores profundicen en el hallazgo, como tratar de explicar por qué los estudiantes de primer año son los más dedicados y vigorosos, en qué pregunta mostró respuestas significativamente diferentes a los otros grados, o por qué el cuarto año es el más bajo. |
Se agregó en conclusiones un párrafo en el cual se profundizó esta materia. |
|
menores: |
|
|
1. Unifique la forma de interpretar los dígitos. |
Se estandarizó la presentación de los dígitos en tablas y textos. |

Reviewer 2 Report
Dear authors, it is very good that you have taken up this important research topic; however, the manuscript requires improvements, especially in the introduction, methodology section.
Introduction
please develop in your introduction the thesis and support it with literature that satisfaction with life is largely determined by a subjective evaluation of one's own assumptions and the fulfilment or not of one's own expectations or, in some cases, of the expectations of others
line 152-161- please complete here the factors that can have an impact on better coping with stress resulting from different situations that concern students
Methodology and Instrument
-Please include in this section a detailed graph showing the selection of the research sample and the course of the experiment
- what inclusion and exclusion criteria were used. It is true that the authors only mention participation in the classes and voluntary participation, but this information is insufficient.
- Please also include information on whether the surveys were voluntary, anonymous and participants were informed about their purpose.
- Please specify how the questionnaires were distributed?
- What measures were taken to reduce errors in the procedure?
- how many questionnaires were distributed, how the data were verified, what was the return rate of the questionnaires and how many questionnaires in total were analysed and how many were rejected and according to what criteria (this is not mentioned anywhere in the paper)
-Were there any limitations to the study, if so, which ones. Would the authors please expand the manuscript to include this paragraph
Author Response
|
Metodología e Instrumento |
|
|
-Incluya en esta sección un gráfico detallado que muestre la selección de la muestra de investigación y el curso del experimento. |
Se agregó una tabla con el detalle de la muestra. |
|
- qué criterios de inclusión y exclusión se utilizaron. Es cierto que los autores sólo mencionan la participación en las clases y la participación voluntaria, pero esta información es insuficiente. |
Se agregó un párrafo en la metodología en que se profundiza respecto a estas materias |
|
- Incluya también información sobre si las encuestas fueron voluntarias, anónimas y si los participantes fueron informados sobre su propósito. |
|
|
- Especifique cómo se distribuyeron los cuestionarios. |
|
|
- ¿Qué medidas se tomaron para reducir los errores en el procedimiento? |
|
|
- cuántos cuestionarios se distribuyeron, cómo se verificaron los datos, cuál fue la tasa de devolución de los cuestionarios y cuántos cuestionarios en total se analizaron y cuántos se rechazaron y según qué criterios (esto no se menciona en ninguna parte del documento) |
|
|
-Hubo limitaciones en el estudio, en caso afirmativo, cuáles. ¿Podrían los autores ampliar el manuscrito para incluir este párrafo? Lo demas se ha mejorado como requerian |
En conclusión hay un párrafo que aborda las limitaciones del estudio. |
